# Crop pest detection by three-scale convolutional neural network with attention

**Xuqi Wang**  *, **Shanwen Zhang, Xianfeng Wang, Cong Xu**

College of Information Engineering, Xijing University, Xi'An, 710123, China

* wangxuqi@xijing.edu.cn

## Abstract

Crop pests seriously affect the yield and quality of crop. To timely and accurately control crop pests is particularly crucial for crop security, quality of life and a stable agricultural economy. Crop pest detection in field is an essential step to control the pests. The existing convolutional neural network (CNN) based pest detection methods are not satisfactory for small pest recognition and detection in field because the pests are various with different colors, shapes and poses. A three-scale CNN with attention (TSCNNA) model is constructed for crop pest detection by adding the channel attention and spatial mechanisms are introduced into CNN. TSCNNA can improve the interest of CNN for pest detection with different sizes under complicated background, and enlarge the receptive field of CNN, so as to improve the accuracy of pest detection. Experiments are carried out on the image set of common crop pests, and the precision is 93.16%, which is 5.1% and 3.7% higher than ICNN and VGG16, respectively. The results show that the proposed method can achieve both high speed and high accuracy of crop pest detection. This proposed method has certain practical significance of real-time crop pest control in the field.

## 1. Introduction

Crop pests not only largely reduce the yield and quality of crop, but also increase the production cost. At present, the main mean to control crop pest is pesticide spraying, which is really effective but has resulted in serious ecological environment destruction. Timely and accurate detection of crop pests is the prerequisite for pest control. At present, there are many crop pest intelligent recognition methods based on crop disease (leaf or fruit) images. Most of them extract artificially set classification features from crop disease images, then train a classifier according to the extracted features, and then use test data for method verification. The effectiveness of these methods largely depends on whether the artificially selected features are reasonable. However, due to the high complexity and diversity of crop disease images, different methods extract different features. Using the existing methods, more than one hundred features can be extracted from a disease image, but it is difficult to determine which features are better. Therefore, these methods have strong subjectivity, low precision and poor generalization ability. With the continuous development of computer, Internet of Things and artificial intelligence technologies, there are many methods of crop pest detection.

**Data Availability Statement:** All relevant data and code can be downloaded free from the following website: https://github.com/xaycq/Small-Pest-dataset.git

**Funding:** This work is supported in the form of grants by the National Natural Science Foundation of China (Nos. 62172338 and 62072378) awarded to SZ, and Xijing University High-level Talent Special Fund Project (No. XJ21B14) awarded to XW.

**Competing interests:** The authors have declared that no competing interests exist.

The detection of crop pest has been a hot topic for many scholars. Martineau et al. [1] introduced and summarized 44 insect classification methods, including image collection, feature extraction, classification and comparison. Yaakob et al. [2] proposed an insect classification method using quality threshold neural network (NN) and six kinds of invariant moments, and used the moments to describe the intra-class and inter-class features of the pest shape. Fedor et al. [3] introduced a semi-automatic tool for insect identification and monitoring based on digital image analysis and artificial NN, in which the quantitative morphological features of insect's head, clavicle, wing, oviposition length and width are extracted and input into NN. Based on the local invariant characteristics, Wen et al. [4] proposed an automatic insect classification method of common insects in orchards. In this method, regional invariant feature detector is used to extract the local features, and scale invariant feature transform (SIFT) descriptors are adopted to represent the pest features and form feature vectors and are input into the classifier. Subsequently, they proposed an image based automatic identification and classification method of crop insects [5]. In the method, a hierarchy combination model is constructed using 54 global features and local features for insect identification and classification. Zhu et al. [6] proposed an automatic insect recognition and classification method based on local average color feature and supported vector machines (SVMs). In the method, each insect image is divided into 40 blocks, and a 480-dimensional eigenvector is extracted to input into SVMs. Wang et al. [7] utilized a series of shape, colour and texture features to develop a content-based image retrieval (CBIR) method for butterfly identification, and compared the experiments with different features, feature weights and similarity matching algorithms. Fina et al. [8] combined K-means clustering algorithm with corresponding filter to realize the detection and identification of crop pests. By extracting the different characteristic attributes between pests and their habitats (leaves and stems), the detection method obtains the relevant peak values of different data sets, and uses the corresponding filter to identify crop pests. Jayme et al. [9] proposed a whitefly identification method based on digital image processing. The method is easily implemented using any image processing software package, and can be easily extended to other crops with little or no algorithm change. Ebrahimi et al. [10] proposed a vision image processing based pest detection method, where mean square error (MSE), root of mean square error (RMSE), mean absolute error (MAE) and mean percent error (MPE) were used for evaluation of the classification. To improve the classification accuracy of insect species in field, Xie et al. [11] developed an insect recognition system using advanced multiple-task sparse representation and multiple-kernel learning (MKL) techniques. Instead of using hand-crafted descriptors, sparse-coding histogram is adopted to represent insect images so that raw features (e.g., color, shape, and texture) can be well quantified.

From the above methods, it is known that these traditional pest detection and recognition algorithms achieved better results, but they have some limitations, such as their detection and recognition performance depend on the prior knowledge to extract and select the appropriate handcraft features designed in advance, which usually include the color, shape and texture of each pest image, while these features are not robust and reliable, and cannot completely distinguish various pests, because the same crop pests in different time may have very different phenotypes, sizes, shapes, postures and positions, as shown in Fig 1. From Fig 1, it is found that the colors, attitudes and shapes of the different pests and even the same pests are various with complex background.

## 2. Related work

Convolutional neural network (CNN) has become dominant in various computer vision tasks, has great advantages in complex image segmentation and feature extraction, is superior to the

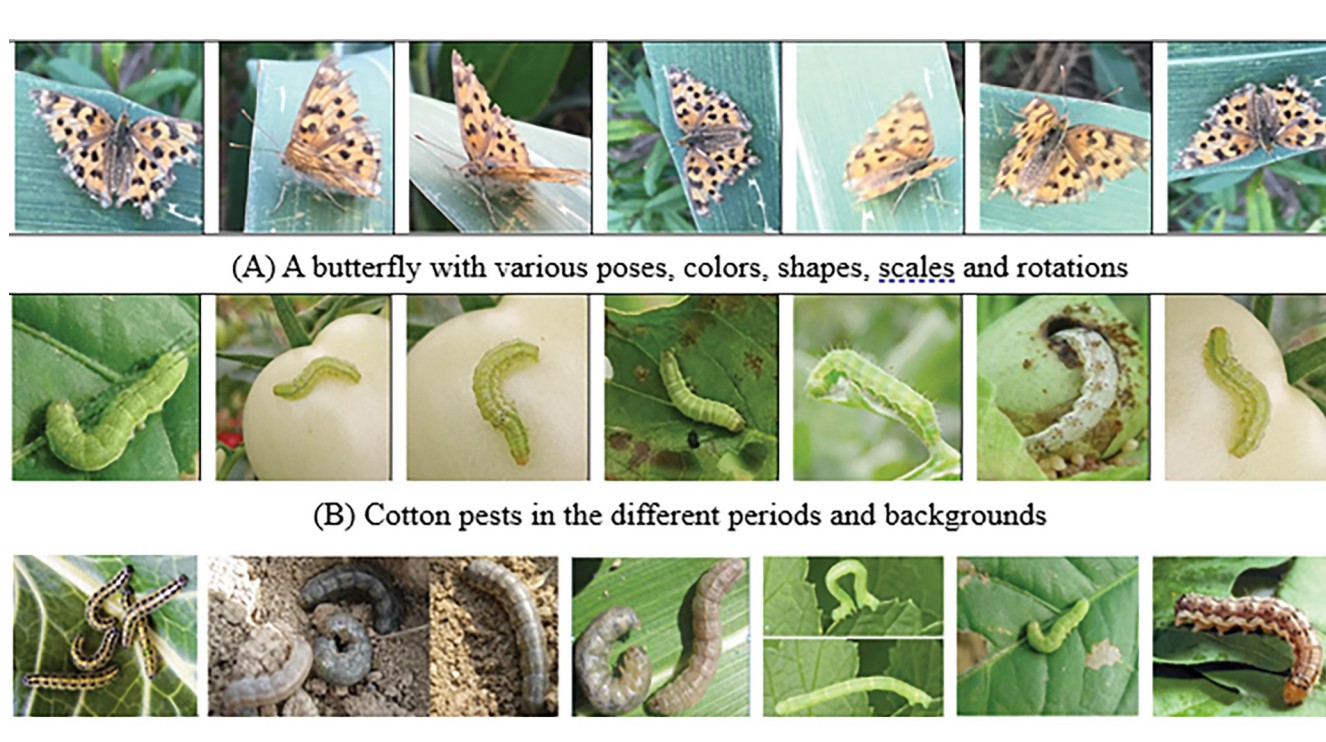

**Fig 1. Crop pest images.** (A) A butterfly with various poses, colors, shapes, scales and rotations. (B) Cotton pests in the different periods and backgrounds. (C) Some kinds of crop pests with various poses, colours, shapes, scales and rotations.

traditional machine learning algorithms in image detection and recognition [12–15], and has been applied to precision agriculture [16,17]. CNN can adjust the weight parameters by itself according to the input training data and its categories to construct the convolutional kernels or appropriate feature extractor, which is relatively efficient and convenient. The constructed feature extractor also has better generalization ability, which can effectively overcome the defects of the traditional methods. CNN can learn multi-level features of images through training, such as patterns, colors and texture features, which makes CNN suitable for pest detection [18]. Kim et al. [19] analyzed unsupervised feature learning using CNN to investigate its efficiency for multi-task classification and compared it to supervised learning features. In agriculture, aphids are one of the most destructive pests in wheat, corn and rape leading to significant economic losses. Li et al. [20] proposed a coarse-to-fine network for aphid detection in dense distribution regions. Improved non-maximum suppression can eliminate overlapping bounding boxes. Bhatt et al. [21] focused on the accuracy performance of the meta-architecture selected by using the data set of small insects, and compared three widely used CNN models for the identification of flying insects. Witenberg et al. [20] proposed a pest recognition method based on an improved residual deep learning model, and presented a pest data set with original images and enhanced images. Xia et al. [22] proposed a CNN model to solve the problem of multiple classifications of crop insects. In order to improve the prediction accuracy and speed up the network training, the regional suggestion network is used instead of the traditional selective search technology to generate fewer suggestion Windows.

From the above analysis, CNN and its improved models have been successfully applied to crop pest detection and recognition, but they are too complex and long training time. In recent years, attention mechanism and multi-scale convolution have been widely applied in various

types of deep learning tasks, such as natural language processing, image recognition and speech recognition [23,24]. The attention mechanism in CNN comes from the biological system of human beings. For example, the visual processing system tends to selectively focus on certain parts of an image and ignore other irrelevant information, thus helping humans to perceive. In text translation and summary tasks, only certain words in the input sequence may be relevant to the next predicted output value. Ling et al. [25] proposed an attention-based CNN (ACNN) for discriminative face feature embedding. ACNN consists of a channel attention block and a spatial attention block. It focuses on the most informative components in both channel and spatial domains to learn the inter-channel relationship matrix and the inter-spatial relationship matrix. Li et al. [26] proposed a multi-branch CNN with attention (MCNNA) to extract the effective features, where its first part is an attention block to reduce the influence of background, its latter part is multi-branch CNN to extract the multi-view features through multi-channel. Zhu et al. [27] presented a two-way attention model using the deep CNN. The model has family first attention and max-sum attention to focus on the discriminative features of the input image. By CNN, the discriminant feature maps of the input images can be extracted through convolutional and pooling operations, but the importance of these feature maps is not consistent in the crop pest detection and recognition. Inspired by CNN and attention mechanism, a three-scale CNN with attention (TSCNNA) model is proposed for multi-scale pest detection, and the experimental results show the effectiveness of the proposed method. The main contributions are summarized as follows:

(1) Multi-scale convolution is used to extract the multi-scale feature for multi-scale pest detection, so that TSCNNA can adapt to pest images of different resolutions.

(2) Attention mechanism is used to enable TSCNNA to focus on crop pests and reduce the impact of background factors, and reduce information loss and accelerate network training.

(3) A large number of experiments have been carried out on the pest dataset.

The rest of this paper is organized as follows. Section 2 simply introduces CNN. TSCNNA is proposed for pest detection in detail in Section 3. Experiments and analysis are conducted in Section 4. Section 5 concludes the paper and gives the future work.

## 3. Convolutional neural network (CNN)

Convolutional neural network (CNN) outperforms the traditional neural network (NN) in two characteristics, local connection and weights sharing, which can improve the network ability, reduce the number of parameters and prevent overfitting. A simple CNN model has several convolutional layers and pooling layers alternately, followed by one or three full connected layers. The architecture of the classical CNN is shown in Fig 2.

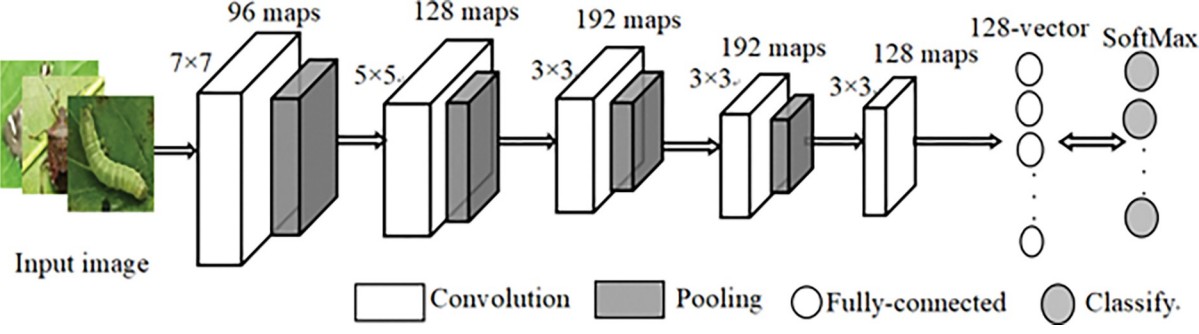

**Fig 2. The architecture of CNN.**

CNN has two outstanding characteristics, local connection (receptive field) and weight sharing. The local connection means that the nodes of the convolutional layer are only connected with part of the nodes of the preceding layer to learn the local features of the image. It can greatly reduce the number of CNN parameters to speed up the model training and learning process, and avoid the overfitting problem to some extent. The weight sharing means that given an input image, the image is convolved with a convolutional kernel. The values in the convolution kernel are called weights. Every position of the image is scanned by the same convolutional kernel, that is, the weights used in the convolution are the same. The weights of a set of convolutional kernels can only obtain a feature map. To better represent the image features, a lot of different convolutional kernels need to be used to enrich the learned image features. Each layer has multiple feature maps, then each feature map has many neurons, each convolutional kernel focuses on a kind of characteristic of the input image, such as the vertical edge, the horizontal edge, color, texture, etc.

The convolutional operation is used to extracted the discriminant feature, which is calculated as follows,

$$z_j^l = \sum_{i \in M_j} x_i^{l-1} * W_{ij}^l + b_j^l \tag{1}$$

where $z_j^l$ is the output of the $i$th neuron of the $l$th convolutional layer, '*' is convolutional operator, $x_i^{l-1}$ is the $i$th input feature in $M_j$ of the $l$th convolutional layer, $M_j$ is the set of input maps in the ($j$-1)th receptive field, $W_{ij}^l$ and $b_j^l$ are the $i$th weight and additive bias of the $j$th convolutional kernel in the $l$th layer, respectively.

The pooling operation is used to reduce the dimension of the feature representation and computational cost in CNN, so as to quickly obtain the spatial invariant features. Max-pooling and average-pooling are often used to avoid the overfitting of CNNs. In the $l$th pooling layer, the output feature on the $j$th local receptive field can be calculated as follows:

$$x_j^l = g(\gamma_j^l down(x_j^{l-1}) + b_j^l) \tag{2}$$

where $g$ is a pooling function, $down(\bullet)$ is the pooling operator, and $\gamma$ is a weight.

The characteristics of the input image are extracted using the convolutional and pooling operators, and then are input into the first fully-connected layer, and the output of the last fully-connected layer is fed to the output layer for image recognition and classification. Let the input of the fully-connected layer be $x$ with the size of $k \times l$, which results in a matrix $Q_{k \times l}$, then the output of CNN is as follows,

$$M(x) = g(Q * x) \tag{3}$$

where $g$ is an activation function, which is often chosen as an identity function.

The Softmax classifier is used to classify the original images into the predefined classes by the feature vector of the last fully-connected layer. The output is a probability vector to indicate the input image belonging to a given class,

$$p_j^{(i)} = \exp(W_j^T x^{(i)} + a_j) / \sum_{n=1}^{C} \exp(W_n^T x^{(i)} + a_n) \tag{4}$$

where $p_j^{(i)}$ is the probability of the sample $x^{(i)}$ belonging to the $j$th class image, $W_j$ and $a_j$ are the $j$th classifier parameter, and $C$ is the number of the predicted image classes.

## 4. Three-scale CNN with attention for pest detection

In the pest image based pest detection and recognition task, some local information may be more closely related to the pest image area. The attention mechanism integrates this relationship, allowing the network model to dynamically focus on specific parts of the input to accomplish the task at hand more effectively. Attentions are generally divided into channel attention and spatial attention. Channel attention is used to select channels to represent different feature information of the image, while spatial attention is applied to select the area of attention in the image feature diagram. To improve the detection accuracy of crop pests, inspired by the multi-scale CNN with attention [25], a three-scale CNN with attention (TSCNNA) model is constructed by introducing the channel and spatial attention into VGG16 [27,28]. Its architecture is shown in Fig 3.

1. Three convolution kernels with sizes of 3×3, 5×5 and 7×7 are used to extract the feature maps at three scales from the original pest images, and are integrated as $F_1 = \max(0, \text{concat}(F_{11}, F_{12}, F_{13}))$ by the integration method [27], where $F_{11}, F_{12}, F_{13}$ are three feature maps obtained by three convolutional kernels respectively, 'concat' is the concatenation operation. The next steps are convolution, pooling, and convolution.

2. In the attention mechanism module, the global average pooling and the global maximum pooling of each feature map are calculated along the channel dimension respectively, and then the channel attention parameters are added to obtain the feature map $F$. Then through the channel attention $M_C$ and spatial attention $M_S$, then $F'$ and $F''$ can be calculated by the following Eq (5).

3. A residual network module is introduced between the 3<sup>th</sup> and 4<sup>th</sup> convolutional layers. The module is used to complement the loss of information and solve the phenomenon of feature

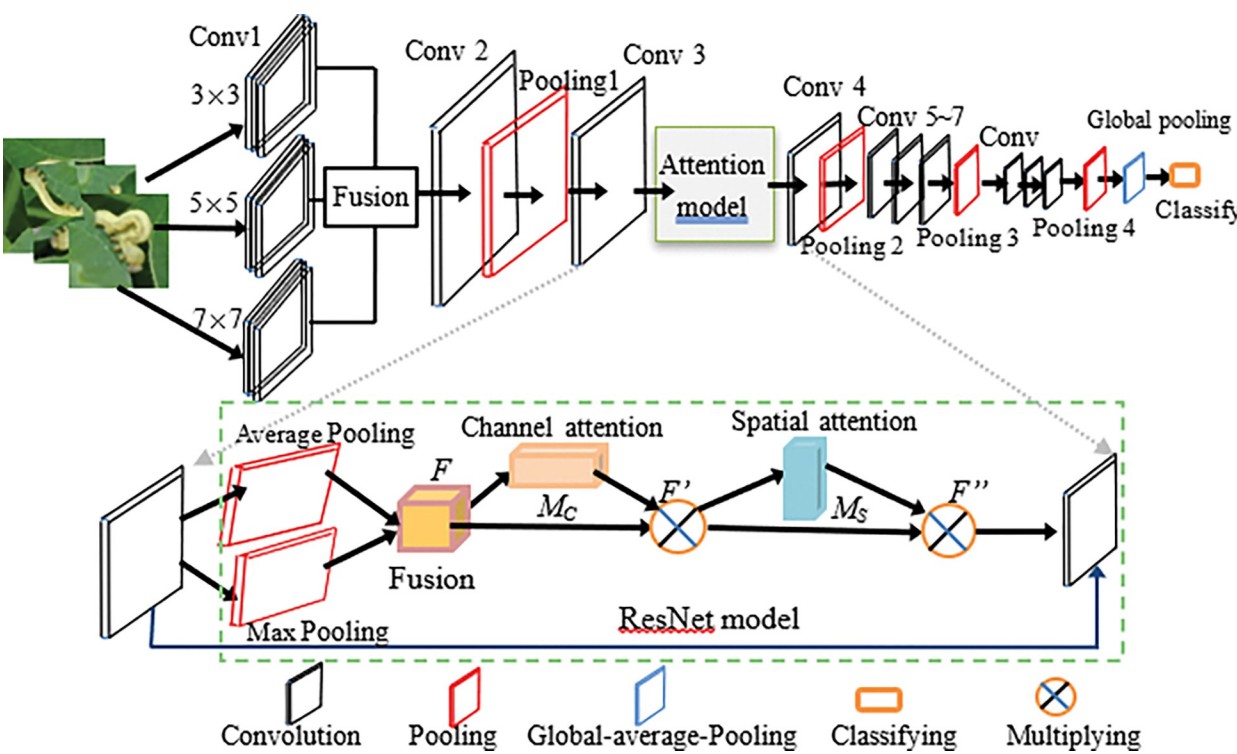

**Fig 3. Architecture of TSCNNA.**

information loss in the process of information transmission and further enhance the feature fusion.

4. The last steps after the fourth convolutional layer is basically the same as VGG16 [22,28], while the three full connection layers of VGG16 are replaced by a global average pooling layer to greatly reduce network parameters.

Given the feature map in the 3th convolutional layer $F \in \mathbb{R}^{H \times W \times C}$, where $H$, $W$ and $C$ represent the length, width and number of channels of the feature map, the channel attention map $M_C$ and spatial attention map $M_S$ are obtained respectively after passing through the channel and spatial convolution attention modules as follows,

$$F' = M_C(F) \otimes F$$
$$F'' = M_S(F') \otimes F' \tag{5}$$

where '$\otimes$' is multiplication for adaptive learning of features, $M_C$ and $M_S$ are calculated as follows,

$$M_C(F) = \sigma(MLP(Avgpool(F)) + MLP(Maxpool(F))) \tag{6}$$

$$M_S(F) = \sigma(f^{7 \times 7}([Avgpool(F); Maxpool(F)])) \tag{7}$$

where $f^{7 \times 7}$ is a convolutional operation of size 7×7, and $\sigma$ is Sigmoid function, $Avgpool(F)$ and $Maxpool(F)$ are average pooling and maximum pooling operations, calculated by

$$Avgpool(F) = \frac{1}{H \times W} \sum_{i=1}^{H} \sum_{j=1}^{W} x_c(i,j) \tag{8}$$

$$Maxpool(F) = argmax \sum_{i=1}^{H} \sum_{j=1}^{W} x_c(i,j) \tag{9}$$

where $X = [x_1, x_2, \ldots, x_n]$ is a feature map, $x_c$ is the $c$-th grayscale value.

Suppose $X$ and $X'$ are two feature maps obtained by the third and fourth convolutional layers, the output is as follows

$$f = X + X' + G'' \tag{10}$$

where $G'' = g(X \odot X' + \xi)$, $\odot$ is element multiplication between matrices, $\xi$ is the tuning offset parameter, $g(\bullet)$ is the differentiable function.

The improved residual connection is calculated as follows:

$$f = X + X' + \sqrt{ReLU(X) \odot ReLU(X') + \xi} \tag{11}$$

The offset of $g(\bullet)$ is set as $\xi = 0.0001$ to maintain gradient stability during back propagation. After the attention and residual operations, the results are expressed as:

$$F''' = F'' + f \tag{12}$$

Since the elements in the matrix and the eigenmatrix may be negative, non-negative processing is required before the calculation. To fuse multilayer convolution features effectively, bilinear interpolation is used to adjust all feature maps to fixed size.

The pest detection is implemented by the SoftMax classifier. Its objective function is formulated as follows:

$$J(W) = -\frac{1}{N}\sum_{n=1}^{N}\sum_{c=1}^{C} \quad (y_n == c)\log\frac{\exp(W_c^T X_i)}{\sum_{p=1}^{C}\exp(W_p^T X_i)} \tag{13}$$

where $(X_i, y_i)(i = 1,2,\ldots,N)$ is the training set, $X_i$ is $i$-th training sample and $y_i$ is the corresponding label. $N$ and $C$ are the numbers of the training samples and classes, and $\ell(*)$ is an indicator function.

The main steps of the pest detection method based on TSCNNA are as follows: (1) Scan and intercept images with the sliding window of different scales to find the candidate Windows that may contain pests in the images to be detected; (2) Normalize the image in the candidate window to 32×32 pixels; (3) The normalized image was used as the input of TSCNNA after training to calculate the network output; (4) Delete the candidate window in the step (3) that is judged to be non-pests, and the remaining candidate window is the position information of pests in the image.

## 5. Experiments and analysis

To verify the proposed method, a lot of experiments are conducted on the actual pest image database, and compared with five pest detection methods, i.e., corn pest detection and identification based on Local features and neural network (LFNN) [4], Local invariant Features and Supported Vector Machines (LFSVM) [6], and Deep learning model (DL) [21], Improved CNN (ICNN) [22], and VGG16 (VGG16) [28]. LFNN and LFSVM are two traditional pest detection methods, while DL, ICNN and VGG16 are three pest detection methods based on deep learning. All experiments are implemented under the operating system Win 10 64bit, processor Intel Xeon E5-2643v3@3.40ghz CPU, 64GB memory, NVidia Quadro M4000 GPU, 8GB video memory, CUDA Toolkit 9.0, CUDNN V7.0, Python version 3.6.4, Tensorflow-GPU 1.8.0 framework [14,26].

### 5.1 Data collection and augmentation

Corn borer, Cabbage night moth, Moth larvae and Cabbage pest are four kinds of common pests in crop production (https://pan.baidu.com/s/1kWEPJDei_fGkPmDAK2OoVA?pwd=p45y). All pest images were collected under the natural background of the field, which laid a foundation for practical application in the later stage. Image acquisition devices include iPhone7, huawei P10, WIFI control camera and Internet of things. An original insect image data set was constructed, containing 1000 insect images of different sizes, 250 images of each insect, with a resolution of about 4928×3264 pixels. Since the iPhone typically captures sharper video, most of the pest video images in the data set were captured by the iPhone7. The image of some crop pests is shown in Fig 4. As can be seen from Fig 4, the shape, color, size, posture, location and background environment of the same pest or even the same pest are diverse, and the size of the pest is relatively small in the image.

The scale of training dataset has great influence on the performance of TSCNNA. When the dimension of the feature space of training samples is larger than the number of the training samples, the model is prone to overfitting. In pest detection, it is difficult and costly to obtain many labeled samples. In order to enhance the robustness and generalization ability of TSCNNA, the limited training set is generally augmented to increase the number of training samples. The commonly used expansion methods include image translation, image rotation,

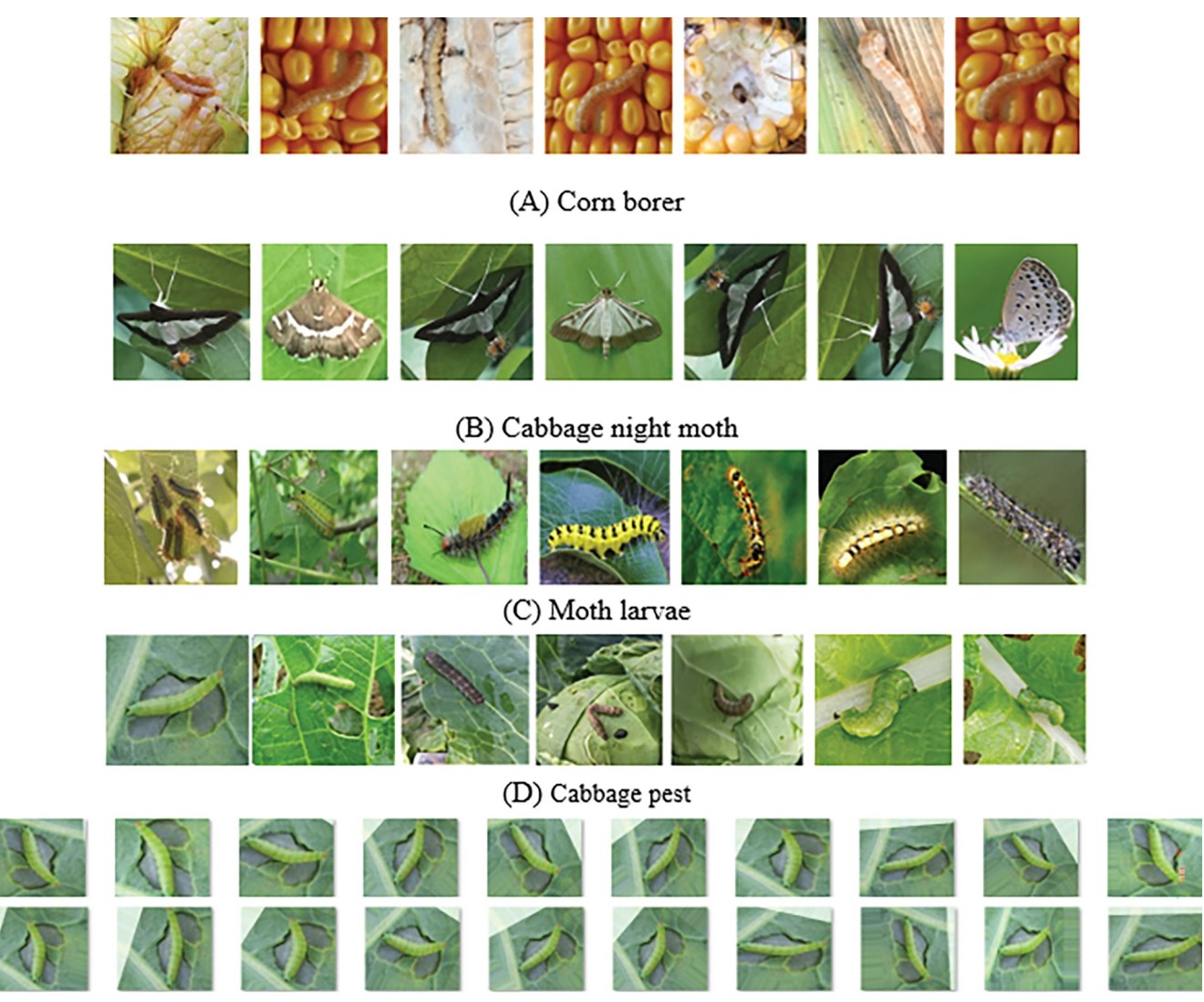

**Fig 4. Four kinds of crop pest images.** (A) Corn borer. (B) Cabbage night moth. (C) Moth larvae. (D) Cabbage pest. (E) 20 augmented images of the first cabbage pest image.

image mirroring, image blur, image brightness change, image clipping, and image zooming. Using the above expansion method, the augmented dataset contains 20,000 images and 5,000 images per pest species. Fig 4E shows 20 extended images of an pest image. Although these pest images have some errors and poor image quality, they can enhance the robustness and generalization ability of the network. Finally, a dataset containing 21 000 images of pests was constructed.

## 5.2 Experiment set

The labelImg tool is used to mark training images for deep learning training (https://github.com/topics/labelimg-tool). This experiment strategy is repeated 100 times, and the stable and reliable average detection result is obtained. In the experiment, 35 images are used for network training in batch, the maximum number of iteration is 10,000, the weight attenuation index is

set as 0.0001 to 0.1, and the initial learning rate is set as 0.01. In the case of 1000, 1500, 2000, 2500 and 2900 iterations, the network training is reduced to 0.0001 through exponential attenuation. The weight attenuation is introduced to reduce the weight attenuation to a smaller value so as to alleviate the overfitting problem. The weight attenuation is 1e-4 and the momentum is 0.9. In the comparison experiment, the parameters are set to the appropriate values. During the test, the category of insect image pixels is determined by the majority vote of the category labels obtained on the overlapping patches. The samples are randomly divided into 5 equal parts by using the five-fold cross-validation method. One part of the test subset and the rest of the training subset are selected in turn. The cross validation is repeated 5 times, and the average value of the cross validation results of 5 times are taken as the final precision of TSCNNA.

In order to analyze the training performance, three network models used for comparison are trained with the same experimental parameters to ensure the reliability of the comparison results. Fig 5(A) and 5(B) show the losses and precisions of TSCNNA and VGG16 versus the number of iterations. It can be seen from Fig 5(A) that TSCNNA is superior to VGG16. Before 1500 iterations, the training process of TSCNNA and VGG16 is relatively stable, while after 1500 iterations, VGG16 drops much slower than TSCNNA. From Fig 5(B), the precision of TSCNNA is higher than that of VGG16 on the whole. The results show that the convergence of TSCNNA is better than that of VGG16, and the training performance of TSCNNA is greatly improved after 2500 iterations. The main reason is that the attention mechanism and residual network are utilized in TSCNNA.

## 5.3 Experiment results

During the training process, the pest training image is input into TSCNNA, and the feature map is gradually extracted through several convolutional layers and pooled layers. In order to test the detection performance of TSCNNA, an image of cabbage pest is randomly selected from the data set to visualize the detection results, as shown in Fig 6. It can be found from the visual feature map in Fig 6 that TSCNNA can capture the fine details of the image of pests, the contour of pests can be extracted with the shallow convolution kernel, and the texture and detailed features of pests can be extracted with the deep convolution kernel. It can be seen from Fig 6E that the convolutional feature map has no obvious sharpening edge and gradually fades, because attention mechanism is introduced into the model to pay more attention to the pest image area rather than the edge of the pest image. As can be seen from Fig 6F, the first feature map is close to the output layer, and the pest image area is more concentrated.

As can be seen from Fig 6, the low-level convolution feature contains more detailed information of pests, while the high-level convolution feature contains the key information of pests. The convolution kernel of the first layer is relatively large, which contains sparse positive and negative response planes, thus increasing the chance of containing appropriate features. The convolution kernel of the second layer is relatively small, easy to train, low gradient and ideal in terms of capacity. Further analysis shows that in the shallow convolutional block, a large number of feature maps are affected by background features, while only a small number can get pest feature maps well. With the deepening of the hierarchy, the pests are better activated and clear, and the irrelevant background can be ignored to extract the valuable high-dimensional classification characteristics of the pests. At the same time, the feature map of deep convolution block can separate pests, and the activation value of the point in the feature map is much brighter than the background. It can also be observed from Fig 6 that the low-level features contain some non-pest information, while the background information of the low-level convolution features is scattered, which leads to the adoption of some measures for the low-

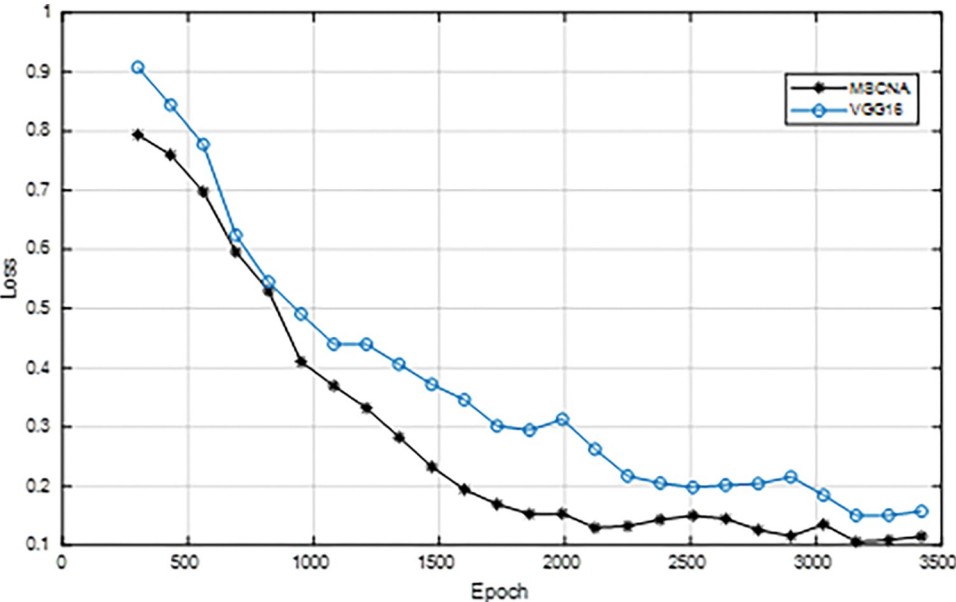

## (A) The loss versus the number of iterations

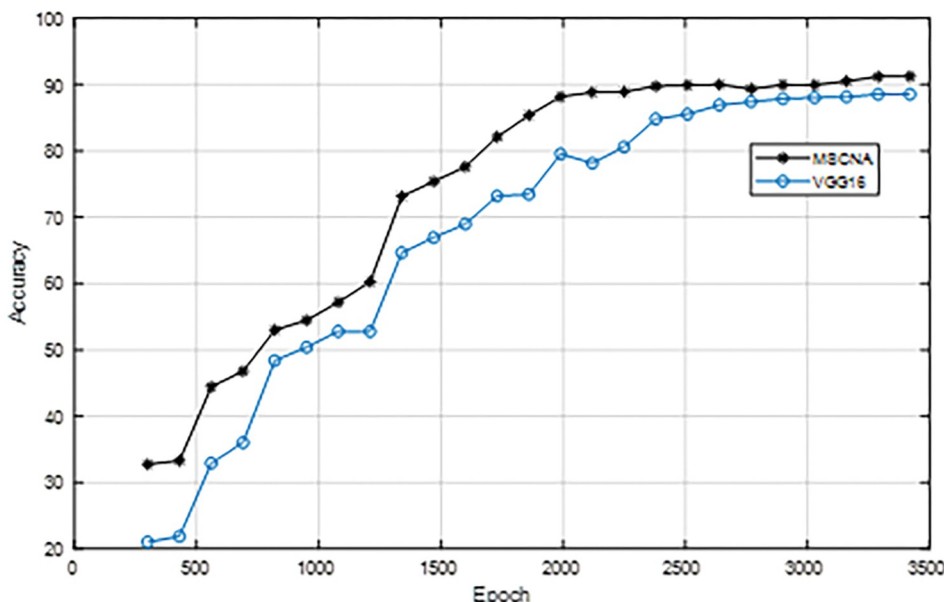

## (B) The precision versus the number of iterations

**Fig 5. The loss and precision versus the number of iterations of TSCNNA and VGG16.** (A) The loss versus the number of iterations. (B) The precision versus the number of iterations.

level convolution features, and it is difficult to directly remove the information of pests. Therefore, in order to effectively connect the low-level convolution feature with the high-level convolution feature and obtain the convolution feature with more discriminative ability, we use the attention mechanism to select the pest feature and remove the non-pest information of the low-level convolution feature. Fig 7 and Table 1 show the detection results of four pests by

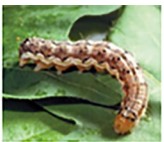

(A) Original Cabbage pest image

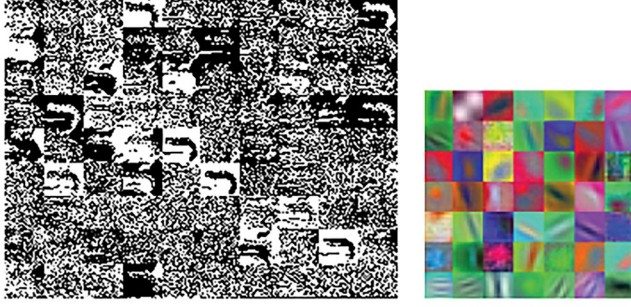

(B) Feature maps and convolutional kernels of the first convolutional layer

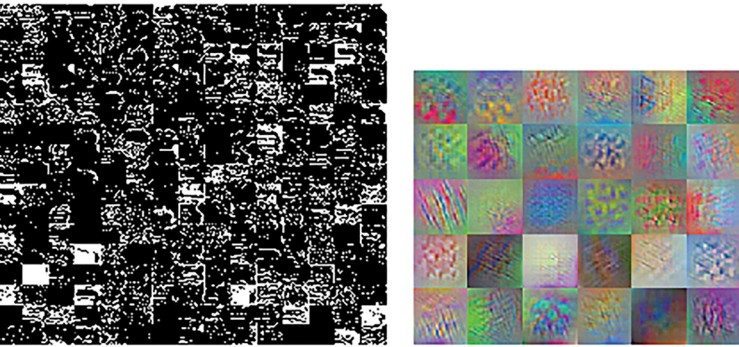

(C) Feature maps and convolutional kernels of the second convolutional layer

(D) Feature maps of the $3^{th}$ convolutional layer          (E) Feature maps of the $4^{th}$ convolutional layer

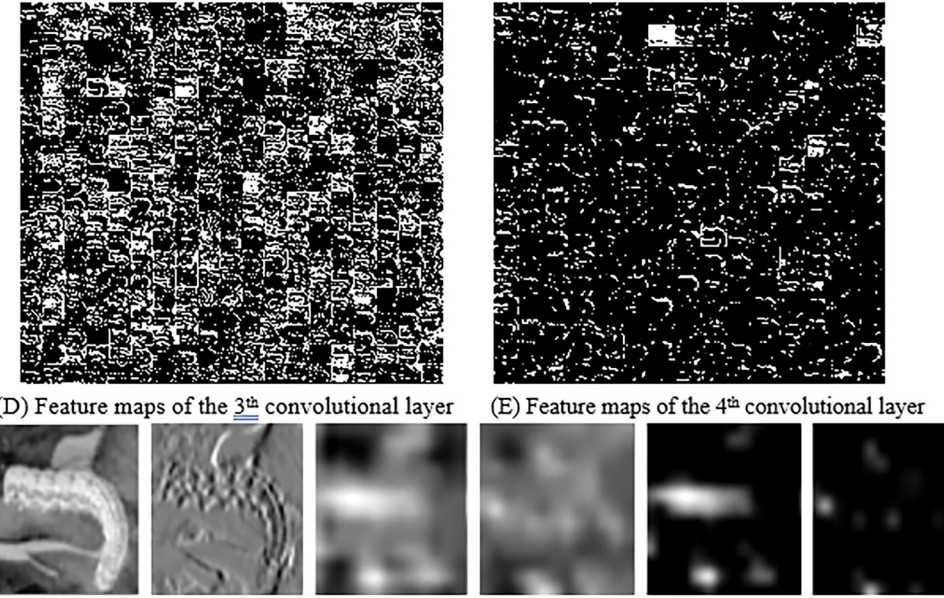

(F) The feature maps of the different convolutional and pooling layers

**Fig 6. The feature maps of the different convolutional layers.** (A) Original Cabbage pest image. (B) Feature maps and convolutional kernels of the first convolutional layer. (C) Feature maps and convolutional kernels of the second

convolutional layer. (D) Feature maps of the 3<sup>th</sup> convolutional layer. (E) Feature maps of the 4<sup>th</sup> convolutional layer. (F) The feature maps of the different convolutional and pooling layers.

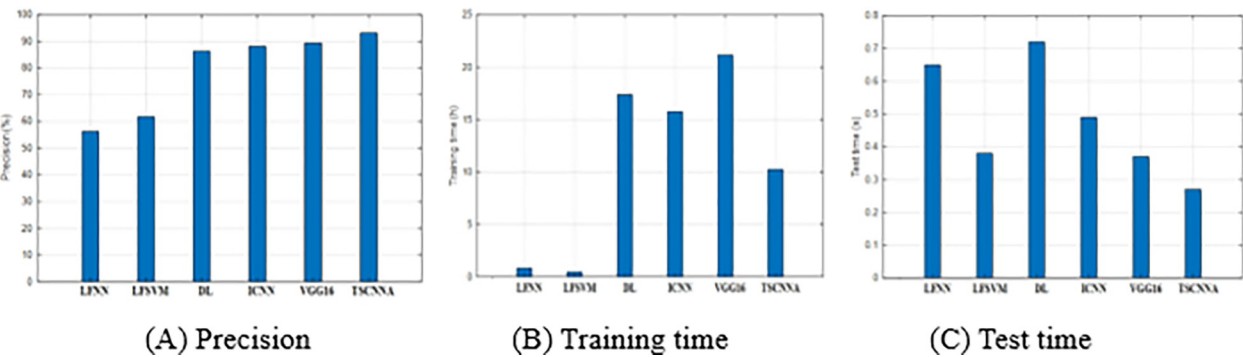

**Fig 7. The detection results of four pest species by six algorithms.** (A) Precision. (B) Training time. (C) Test time.

**Table 1. The detection results of four pest species by six algorithms.**

| Method <br> Results | LFNN | LFSVM | DL | ICNN | VGG16 | TSCNNA |
|---|---|---|---|---|---|---|
| Precision (%) | 56.25 | 61.76 | 86.34 | 88.15 | 89.37 | 93.16 |
| Training time (h) | 0.84 | 0.47 | 17.36 | 15.74 | 21.16 | 10.24 |
| Test time (s) | 0.65 | 0.38 | 0.72 | 0.49 | 0.37 | 0.27 |

LFNN [4], LFSVM [6], DL [21], ICNN [22], VGG16 [28] and TSCNNA proposed in this paper for the images.

It can be seen from Fig 7 and Table 1 that: (1) the precision of the three DL-based methods is much higher than that of the two traditional methods. The reason is that the DL models can automatically learn the classification features of images from complex images, while the traditional methods are difficult to extract the robust features of pests, because a pest in the field varies from time to time, as shown in Fig 1. Generally, the traditional methods require preprocessing and segmentation of pest images. All six methods in Table 1 use the original image and its extended image for experiments. Since the two traditional methods cannot extract effective classification features, their recognitions are very low. (2) The training time of the three DL-based methods is much more than that of the two traditional methods. The reason is that compared with NN and SVM, the DL model needs to spend a lot of time to train the parameters of the model. (3) For training NN, SVM and DL models, there is little difference in test time. (4) The proposed method is superior to other methods, with the highest detection rate and the least training and detection time.

To demonstrate the detection performance at different dataset scales, a large number of experiments are conducted and adopted a five-fold cross-validation scheme with different numbers of pest images. Under different number of training samples, the detection results of the proposed method are shown in Fig 8. It can be seen from Fig 8 that the more pest images, the higher the detection accuracy. When the number of training samples is less than 6000, the detection accuracy improves rapidly, when the number of training samples is more than 11,000, the detection accuracy improves stably, while the training samples over 16,000 do not further improve the detection accuracy. The reason is that when the training samples are not

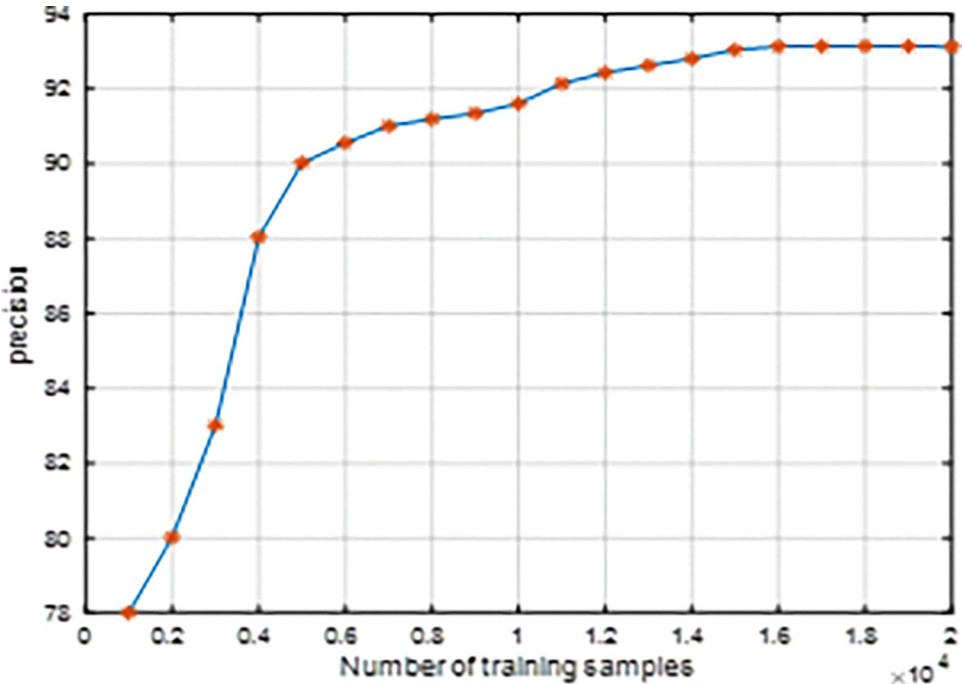

**Fig 8. The precision versus the number of the training samples.**

enough, the proposed method cannot represent various pest image features, so the detection accuracy is not high. But, when the training samples are large enough, if the training samples are increased, the model TSCNNA training tends to converge, so the detection accuracy will not further increase.

In general, the TSCNNNA based method performs supervised learning on the basis of a small amount of data, avoids complex image processing process, and is better than other methods in terms of accuracy, training and test time. The precision is more than 91%, indicating that TSCNNA has a very good classification performance. The main reason is that multi-scale and attention mechanism are introduced into TSCNNA. In the feature extraction stage, the weighted feature vector after convolution is used to replace the original feature vector for residual fusion, and the second-order term is used to reduce information loss and speed up network training in the training process.

## 6. Conclusions

Timely and accurate detection of crop pests is the premise of precise control of crop pests. Crop pest detection is an important and challenging research direction due to the variety of field pests, the variety of postures and forms of the same pest, and the complex field background. Combining attention mechanism, residual model and VGG16, a TSCNNA model was proposed and applied to multi-scale pest detection. In this model, channel attention and spatial attention are introduced in the feature extraction process, and then the filtered weighted feature vector is used to replace the original feature vector, and then the residual fusion is carried out. Finally, the second-order residual term is introduced in the training process, so as to reduce information loss and speed up the network training speed. The experimental results on the pest image data set show that TSCNNA is an effective pest detection model, and its detection accuracy reaches more than 90%. The pest images in the experiment were collected under

field conditions rather than under ideal conditions, which makes the proposed model have strong anti-interference ability. The pest recognition method based on TSCNNA can locate the pests in the image, while the other three methods can realize the image classification of pests. Therefore, the method in this paper can effectively alleviate the interference and burden of human factors in data processing. This method can be applied to automatic detection and identification system of crop pests. Future work is as follows: (1) the pest database needs to be expanded, which is not enough to reflect the feasibility of the proposed method. In the later stage, the Internet of things is used to automatically collect pest images and expand the data set; (2) In the process of pest control, the classification of pests needs to be more detailed, the growth cycle of pests needs to be divided, and the categories of pests need to be identified according to the different growth periods of pests, so that different control measures can be taken.

## Author Contributions

**Data curation:** Shanwen Zhang.

**Formal analysis:** Xianfeng Wang.

**Methodology:** Xuqi Wang, Shanwen Zhang, Xianfeng Wang.

**Resources:** Xianfeng Wang.

**Software:** Xuqi Wang.

**Visualization:** Cong Xu.

**Writing – original draft:** Xuqi Wang.

**Writing – review & editing:** Shanwen Zhang, Cong Xu.

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
