## [Decision Letter · Decision Letter 0]

26 Aug 2022

PONE-D-22-22643Crop Pest Detection by Three-scale Convolutional Neural Network with AttentionPLOS ONE

Dear Dr. Wang,

Thank you for submitting your manuscript to PLOS ONE. After careful consideration, we feel that it has merit but does not fully meet PLOS ONE’s publication criteria as it currently stands. Therefore, we invite you to submit a revised version of the manuscript that addresses the points raised during the review process.

We look forward to receiving your revised manuscript.

Kind regards,

Kapil Kumar Nagwanshi, PhD

Academic Editor

PLOS ONE

Reviewers' comments:

Reviewer's Responses to Questions

**Comments to the Author**

1. Is the manuscript technically sound, and do the data support the conclusions?

Reviewer #1: Yes

Reviewer #2: Yes

2. Has the statistical analysis been performed appropriately and rigorously? 

Reviewer #1: Yes

Reviewer #2: Yes

3. Have the authors made all data underlying the findings in their manuscript fully available?

Reviewer #1: Yes

Reviewer #2: Yes

4. Is the manuscript presented in an intelligible fashion and written in standard English?

Reviewer #1: Yes

Reviewer #2: Yes

5. Review Comments to the Author

Reviewer #1: In submitted manuscript Author(s) proposed 'Crop Pest Detection by three scale Convolutional Neural Network with Attention' which will help crop security, life and stable agricultural economy. Here are few comments related to submitted manuscript. Submitted manuscript is written in well structured manner and it has a great impact on social life as well as farming industries.

1. In abstract, author(s) must show the shows the comparative results in terms of existing method and proposed method.

2. In Experiments and analysis section author (s) use so many key words (such as Toolkit 9.0, CUDNN V7.0, PYTHON 3.6.4, TENSORFLOW-GPU 1.8.0 etc), add citation for each key words and also add in reference section.

3. In section 4.2 author(s) says that ' the labeling tool is used..........' at citation for 'Labeling tool in revised manuscript and add in reference section.

4. Data shows in table-1, must represented using bar graph, so that performance of proposed method can easily compaired in terms of precision, Training time and Test time.

5. Size of data set 1000 is less what happened if data set increase, what % is used for training and test your proposed method in data set.

6. Is the data set used in the submitted manuscript available for researchers if yes provide link if no why?

7. Important references are missing for CNN, ML and VGG, Introduction of each is required in introduction section. some references are given below which must included in revised manuscript related to CNN, ML and VGG.

-Aquila coyote-tuned deep convolutional neural network for the classification of bare skinned images in websites

-Cohort study on recognition of plant species using Deep Learning methods

-Deep Learning (CNN) and Transfer Learning: A Review

- A Systematic Literature Review of Breast Cancer Diagnosis Using Machine Intelligence Techniques

8. Is any software implementations available for public use for the proposed algorithms you have proposed, If Yes give the link in revised manuscript, if no why?

9. It is better to divide section-1 into two separate section 1 Introduction 2. Literature Revie.

Reviewer #2: 1. Poor literature survey. Authors can add a separate section of the “Related Works” to investigate the recent published work for proposing the approach.

2. Author stated that “existing convolutional neural network (CNN) based pest detection methods are not satisfactory for small pest recognition and detection in field”. Give the justification with taking appropriate references. What does mean “small pest recognition and detection in field”.

3. Write Fig. 5 instead of Fig. 5AB. It is suggested to proofread completely

4. Contributions can be enhanced, to inform readers the contributions made in advance.

6. PLOS authors have the option to publish the peer review history of their article (what does this mean?). If published, this will include your full peer review and any attached files.

Reviewer #1: No

Reviewer #2: No

---

## [Author Response · Author response to Decision Letter 0]

18 Sep 2022

In the following, let us state the changes to the manuscript and the replies to the reviewers.

Responses to reviewers' comments:

Reviewer #1:

1. In abstract, author(s) must show the comparative results in terms of existing method and proposed method.

Response: Thanks for your valuable suggestion. 

Experiments are carried out on the image set of common crop pests, and the recognition rate is 93.16%, which is 5.1% and 3.7% higher than ICNN and VGG16, respectively. 

2. In Experiments and analysis section author (s) use so many key words (such as Toolkit 9.0, CUDNN V7.0, PYTHON 3.6.4, TENSORFLOW-GPU 1.8.0 etc.), add citation for each key words and also add in reference section.

Response: Thanks for your valuable suggestion. In various papers, the software and hardware conditions of training and testing deep learning models are abnormal. Our software environment, similar to Ref. [14,26], is built as follows: the operating system Win 10 64bit, processor Intel Xeon E5-2643v3@3.40ghz CPU, 64GB memory, NVidia Quadro M4000 GPU, 8GB video memory, CUDA Toolkit 9.0, CUDNN V7.0, Python version 3.6.4, Tensorflow-GPU 1.8.0 framework [14,26].

3. In section 4.2 author(s) says that ' the labeling tool is used..........' at citation for 'Labeling tool in revised manuscript and add in reference section.

Response: Thanks for your finding. We used labelImg tool for labeling the training samples. Its download site is as follows: 

abelImg tool (https://github.com/topics/labelimg-tool)

4. Data shows in table-1, must represented using bar graph, so that performance of proposed method can easily compared in terms of precision, Training time and Test time.

Thanks for your valuable suggestion. When comparing the data, the table is not obvious and intuitive, and has been revised as required. The revised bar graph is as follows:

Fig. The detection results of four pest species by six algorithms

We found that the detection results of Precision, Training time and Test time are shown in an image is inappropriate, while shown in following three images is suitable. 

 (A) Precision (B) Training time (C) Test time

Fig. The detection results of four pest species by six algorithms

5. Size of data set 1000 is less what happened if data set increase, what % is used for training and test your proposed method in dataset.

Response: Thanks for your finding. We used five-fold cross-validation scheme to conduct the detecting experiments. 

In order to demonstrate the detection performance at different dataset scales, a large number of experiments are conducted and adopted a five-fold cross-validation scheme with different numbers of pest images. Under different number of training samples, the detection results of the proposed method are shown in Fig.8. It can be seen from Fig.8 that the more pest images, the higher the detection accuracy. When the number of training samples is less than 6000, the detection accuracy improves rapidly, when the number of training samples is more than 11,000, the detection accuracy improves stably, while the training samples over 16,000 do not further improve the detection accuracy. The reason is that when the training samples are not enough, the proposed method cannot represent various pest image features, so the detection accuracy is not high. But, when the training samples are large enough, if the training samples are increased, the model TSCNNA training tends to converge, so the detection accuracy will not further increase. 

Fig.8 The precision versus the number of the training samples

6. Is the dataset used in the submitted manuscript available for researchers if yes provide link if no why?

Response: Thanks for your finding. We gave our dataset link for free download: 

https://pan.baidu.com/s/1kWEPJDei_fGkPmDAK2OoVA?pwd=p45y, extraction code: P45Y

7. Important references are missing for CNN, DL and VGG, Introduction of each is required in introduction section. some references are given below which must included in revised manuscript related to CNN, DL and VGG.

Response: Thanks for your finding. We have referenced following 4 references in the revised manuscript related to CNN, DL and VGG.

[12] Barhate D, Pathak S, Dubey A K, et al. Cohort study on recognition of plant species using Deep Learning methods. Journal of Physics: Conference Series. IOP Publishing, 2022, 2273(1): 012006.

[13] Nemade V, Pathak S, Dubey A K. A Systematic Literature Review of Breast Cancer Diagnosis Using Machine Intelligence Techniques. Archives of Computational Methods in Engineering, 2022: 1-30.

[14] Gupta J, Pathak S, Kumar G. Deep Learning (CNN) and Transfer Learning: A Review. Journal of Physics: Conference Series. IOP Publishing, 2022, 2273(1): 012029.

[15] Gupta J, Pathak S, Kumar G. Aquila coyote-tuned deep convolutional neural network for the classification of bare skinned images in websites. International Journal of Machine Learning and Cybernetics, 2022, 13(10): 3239-3254.

8. Is any software implementations available for public use for the proposed algorithms you have proposed, If Yes give the link in revised manuscript, if no why?

Thanks for your valuable suggestion.

Response: Thanks for your finding. We gave our dataset link for free download: 

https://pan.baidu.com/s/1kWEPJDei_fGkPmDAK2OoVA?pwd=p45y, extraction code: P45Y

9. It is better to divide section-1 into two separate section 1 Introduction 2. Literature Review.

Response: Thanks for your finding. We revised as 

1. Introduction 

2. Related work.

Reviewer #2:

1. Poor literature survey. Authors can add a separate section of the “Related Works” to investigate the recent published work for proposing the approach.

Thanks for your valuable suggestion. We added a separate section of the “Related Works” to investigate the recent published work for proposing the approach.

2. Author stated that “existing convolutional neural network (CNN) based pest detection methods are not satisfactory for small pest recognition and detection in field”. Give the justification with taking appropriate references. What does mean “small pest recognition and detection in field”.

Thanks for your valuable finding. Appropriate references are as follows:

Small pests often go undetected because much information is lost during the pooling process. In CNN, after a pooling, the output feature map size is reduced by half and the resolution is reduced by half, so the feature of the small pest image may be submerged [23]. 

[23] Wei D, Chen J, Luo T , et al. Classification of crop pests based on multi-scale feature fusion. Computers and Electronics in Agriculture, 2022, 194: 106736. DoI: 10.3390/insects13060554.

3. Write Fig. 5 instead of Fig. 5AB. It is suggested to proofread completely

Thanks for your valuable finding. We revised Fig. 5 AB as Figs.5 (A) and (B).

4. Contributions can be enhanced, to inform readers the contributions made in advance.

Response: Thanks for your finding. The contributions are summarized as follows.

(1) Multi-scale convolution is used to extract the multi-scale feature for multi-scale pest detection, so that TSCNNA can adapt to pest images of different resolutions. 

(2) Attention mechanism is used to enable TSCNNA to focus on crop pests and reduce the impact of background factors, and reduce information loss and accelerate network training.

Finally, thanks again to the reviewers for their comments. Hopefully, we have addressed your concern. If there is any other problem, please let us know.

---

## [Decision Letter · Decision Letter 1]

7 Oct 2022

Crop Pest Detection by Three-scale Convolutional Neural Network with Attention

PONE-D-22-22643R1

Dear Dr. Wang,

We’re pleased to inform you that your manuscript has been judged scientifically suitable for publication and will be formally accepted for publication once it meets all outstanding technical requirements.

Kind regards,

Kapil Kumar Nagwanshi, PhD

Academic Editor

PLOS ONE

Additional Editor Comments (optional):

Reviewers' comments:

Reviewer's Responses to Questions

**Comments to the Author**

1. If the authors have adequately addressed your comments raised in a previous round of review and you feel that this manuscript is now acceptable for publication, you may indicate that here to bypass the “Comments to the Author” section, enter your conflict of interest statement in the “Confidential to Editor” section, and submit your "Accept" recommendation.

Reviewer #1: All comments have been addressed

Reviewer #2: All comments have been addressed

2. Is the manuscript technically sound, and do the data support the conclusions?

Reviewer #1: Yes

Reviewer #2: Yes

3. Has the statistical analysis been performed appropriately and rigorously? 

Reviewer #1: Yes

Reviewer #2: Yes

4. Have the authors made all data underlying the findings in their manuscript fully available?

Reviewer #1: Yes

Reviewer #2: Yes

5. Is the manuscript presented in an intelligible fashion and written in standard English?

Reviewer #1: Yes

Reviewer #2: Yes

6. Review Comments to the Author

Reviewer #1: NA, all comments are incorporated in revised manuscript now the revised manuscript is ACCEPTED for publication

Reviewer #2: All the comments have been properly addressed by the authors as:

1. Authors added separate section of "Related Works".

2. Authors have performed proofread in order to improve the readability of article.

3. Contributions have incorporated.

No further changes are required.

7. PLOS authors have the option to publish the peer review history of their article (what does this mean?). If published, this will include your full peer review and any attached files.

Reviewer #1: No

Reviewer #2: No

---

## [Editor Report · Acceptance letter]

25 May 2023

PONE-D-22-22643R1 

Crop Pest Detection by Three-scale Convolutional Neural Network with Attention 

Dear Dr. Wang:

I'm pleased to inform you that your manuscript has been deemed suitable for publication in PLOS ONE. Congratulations! Your manuscript is now with our production department. 

Kind regards, 

on behalf of

Dr. Kapil Kumar Nagwanshi 

Academic Editor

PLOS ONE